# Impaired Audiovisual Representation of Phonemes in Children with Developmental Language Disorder

**DOI:** 10.3390/brainsci11040507

**Published:** 2021-04-16

**Authors:** Natalya Kaganovich, Jennifer Schumaker, Sharon Christ

**Affiliations:** 1Department of Speech, Language, and Hearing Sciences, Purdue University, 715 Clinic Drive, West Lafayette, IN 47907-2038, USA; jschumak@purdue.edu; 2Department of Psychological Sciences, Purdue University, 703 Third Street, West Lafayette, IN 47907-2038, USA; 3Department of Statistics, Purdue University, 250 N. University Street, West Lafayette, IN 47907-2066, USA; slchrist@purdue.edu; 4Department of Human Development and Family Studies, Purdue University, 1202 West State Street, West Lafayette, IN 47907-2055, USA

**Keywords:** audiovisual processing, phonemic representations, developmental language disorder, audiovisual development, event-related potentials, mismatch negativity

## Abstract

We examined whether children with developmental language disorder (DLD) differed from their peers with typical development (TD) in the degree to which they encode information about a talker’s mouth shape into long-term phonemic representations. Children watched a talker’s face and listened to rare changes from [i] to [u] or the reverse. In the neutral condition, the talker’s face had a closed mouth throughout. In the audiovisual violation condition, the mouth shape always matched the frequent vowel, even when the rare vowel was played. We hypothesized that in the neutral condition no long-term audiovisual memory traces for speech sounds would be activated. Therefore, the neural response elicited by deviants would reflect only a violation of the observed audiovisual sequence. In contrast, we expected that in the audiovisual violation condition, a long-term memory trace for the speech sound/lip configuration typical for the frequent vowel would be activated. In this condition then, the neural response elicited by rare sound changes would reflect a violation of not only observed audiovisual patterns but also of a long-term memory representation for how a given vowel looks when articulated. Children pressed a response button whenever they saw a talker’s face assume a silly expression. We found that in children with TD, rare auditory changes produced a significant mismatch negativity (MMN) event-related potential (ERP) component over the posterior scalp in the audiovisual violation condition but not in the neutral condition. In children with DLD, no MMN was present in either condition. Rare vowel changes elicited a significant P3 in both groups and conditions, indicating that all children noticed auditory changes. Our results suggest that children with TD, but not children with DLD, incorporate visual information into long-term phonemic representations and detect violations in audiovisual phonemic congruency even when they perform a task that is unrelated to phonemic processing.

## 1. Introduction

Although some ability to match auditory and visual information emerges early in infancy (e.g., [1,2]), other audiovisual skills, such as audiovisual temporal encoding [3] and speech-in-noise perception [4], continue to develop during school years. This protracted development of audiovisual skills makes a lot of sense given that the temporal lobe—the seat of the key components of the audiovisual integration network—is among the last cortical areas to mature [5,6]. Impairments in the development of audiovisual function have now been documented in several neurodevelopmental disorders, such as autism (e.g., [7,8,9,10]), dyslexia [11,12], and developmental language disorder (DLD, also known as specific language impairment or SLI, e.g., [13,14]). The nature of the impairments in each group and their relationship to other aspects of these disorders require a great deal more research; however, at least in some cases, audiovisual deficits appear to contribute critically to key aspects of these disorders (e.g., [15,16]).

In this study, we focused on audiovisual processing in children with DLD. DLD is characterized by impairments in language processing that cannot be easily accounted for by hearing impairment, neurological damage, or low nonverbal intelligence [17]. Children with DLD are a heterogeneous group. Although most profound deficits in this population are often in the area of morphosyntax, phonological and auditory processing deficits are also often present, as well as deficits in working memory and attention [18]. At 7% prevalence [19], DLD affects at least as many children as autism (approximately 1.5%, [20]) and stuttering (approximately 5%, [21]) do combined. DLD is a debilitating disorder, and the prognosis for many children with DLD is poor. While some children’s language skills improve with age and therapy, most are left with varying degrees of life-long language difficulties [22], which interfere with their personal and professional lives [23,24].

Earlier studies showed reduced sensitivity to visual speech cues in children with DLD. For example, they are not influenced by the McGurk illusion (in which, typically, a sound “pa” is dubbed onto the visual articulation of “ka”, leading to the perception of “ta” or “tha”) as strongly as children with typical development (TD) are (e.g., [25,26,27,28]). Congruent audiovisual processing also seems to be atypical in this group. In an earlier study in our laboratory [14], children with DLD were significantly less accurate than their peers with TD in matching auditory words with silent visual articulations. They also benefited less from seeing the talker’s face during the speech-in-noise perception task. The event-related potential (ERP) recordings collected while children were watching videos of articulations revealed that the DLD group had a reduced phonological N400 to lip movements when such movements clearly did not match the preceding auditory words, suggesting that children with DLD do not detect audiovisual correspondences during speech perception as well as children with TD do.

Although earlier studies demonstrate clear audiovisual processing difficulties in children with DLD, the specific cognitive and neural mechanisms underlying them are less clear. For example, the ability to perceive the McGurk illusion varies significantly even among healthy young adults and recruits neural regions that are at least partially distinct from those that are active during perception of the more natural congruent audiovisual speech (e.g., [29,30,31]). Therefore, the degree to which reduced McGurk susceptibility in DLD is clinically significant is difficult to determine. The audiovisual word-matching task used in our earlier study was complex and likely required a range of cognitive skills that are not language-specific, such as the ability to allocate visual attention and to create a prediction for what the speaker’s mouth movements will look like based on the heard word.

In this study, we examined one critical aspect of audiovisual speech perception in children with DLD—namely, the nature of their long-term phonemic representations. Phonological knowledge has been shown to significantly impact other aspects of linguistic development, such as fast mapping during word acquisition [32], morphological development [33], reading [34], and even arithmetic [35]. However, in most cases, phonemic representations in children have been studied in the auditory modality only.

As we summarized earlier, studies in adults show strong associations between auditory and visual components of speech, including at the phonemic level, and suggest ways in which such associations may be facilitative. In typical development, the ability to match mouth shapes with the sounds they can produce emerges as early as 2 months of age [1]. Sensitivity to visual speech information also undergoes a narrowing during the first year of life, becoming more fine-tuned to the child’s native language, similarly to the narrowing observed in the auditory modality [2]. While the ability to match phonetic information in auditory and visual modalities emerges early, it continues to develop well into the mid-teen years and is influenced by task and by how easily one can identify a sound by observing a speaker’s mouth [36,37,38,39]. Previous research suggests that phonemic representations in which visual information is well specified could confer benefits not only for speech-in-noise perception but also for attentional allocation to speech signal and for lexical access, among other things [40,41,42], thus potentially affecting multiple levels of speech and language processing. We, therefore, asked whether children with DLD incorporate visual information into phonemic representations to the same extent as children with TD do. We used an audiovisual oddball paradigm in order to probe the degree of visual influence on phoneme perception in children with TD and DLD while minimizing cognitive demands needed to perform the task.

Our main experimental measure was the amplitude of the mismatch negativity (MMN) event-related potential (ERP) component. The auditory MMN ERP component has been used extensively to study auditory processing in individuals with normal [43,44,45] and impaired (e.g., [46]) neural function. It is elicited by any noticeable change in an auditory pattern. Importantly, MMN reflects not only auditory but also linguistic processing such that the same auditory change may elicit MMN components of different amplitude depending on whether or not this change is linguistically significant (i.e., represents a phonemic, rather than just an acoustic) change in participants’ native language [47,48,49].

Visual MMN shares a lot of characteristics with its auditory counterpart (for a review, see [50]). It is elicited by detectable changes in visual sequences [51,52]. Such changes may contain individual visual features, such as color (e.g., [53]), shape (e.g., [54]), and movement trajectories [55], as well as their conjunctions (e.g., [56]). The amplitude of visual MMN may also be influenced by whether or not observed changes occur within the same visual object [57] and/or represent a category change [58]. One notable difference between auditory and visual MMN components is their scalp distribution, with visual MMN occurring primarily over the posterior scalp. Both components peak approximately 150–250 ms following the detected change, although the latency range reported from different experimental designs for visual MMN is quite broad.

Studies of audiovisual MMN are few. Besle and colleagues were the first to explore audiovisual memory traces in oddball paradigms [59,60]. In their studies, standards and deviants shared the same auditory (two complex tones increasing in frequency) and visual (circles stretching into ovals horizontally or vertically) features but differed in their combinations. Compared to auditory-only changes, audiovisual deviants elicited MMN that was smaller and later and had an additional source in the occipital lobe. A later study by the same group modified the paradigm by varying how consistently auditory and visual stimuli were combined in standards. Auditory-only changes that occurred in blocks with consistent audiovisual standards elicited a larger MMN over frontal scalp compared with blocks with inconsistent audiovisual standards, providing additional support to their earlier conclusion that the neural encoding of audiovisual regularities is possible. However, how well these findings with abstract stimuli reflect the neural processes occurring during audiovisual speech perception is unclear.

In most oddball paradigms, attention is diverted away from the rare change to the opposite modality. For example, it is common for participants in an auditory MMN study to watch a silent movie during testing. In our paradigm, children’s attention was directed to the visual modality, as described below. However, all stimuli were audiovisual. Therefore, we expected that rare stimulus changes might lead to attentional shifts and elicit not only MMN but also a P3 component [44]. To examine whether the degree of such attentional shifts depended on the type of deviants, we have included the P3 component in our analyses.

In this study, children monitored a talker’s face for silly expressions and pressed a response button every time they detected them. Unrelated to this task, we presented two types of phonemic changes. Children heard infrequent changes from vowel [i] to [u] or the reverse. Critically, in some blocks the talker’s lips matched the standard vowel while in others they remained closed and in a neutral position. We hypothesized that in blocks in which lip shape remained neutral, no long-term audiovisual memory traces for speech sounds would be activated. Therefore, the neural response elicited by deviants in these blocks would violate only the audiovisual sequence specific to any given block. In contrast, we expected that in blocks in which the talker’s lip shape matched the frequent vowel, a long-term memory trace for the speech sound/lip configuration typical for that vowel would be activated. In these blocks then, the neural response elicited by infrequent sound changes would reflect a violation of not only block-specific audiovisual patterns (i.e., the repeated lip shape and sound combination) but also of a long-term memory representation for how a given vowel looks when articulated.

Regarding key group comparisons, we hypothesized that if children with DLD differed from their peers with TD in the integration of visual (lip shape) and auditory (heard vowel) components of speech sounds, we should see a reduction in the MMN component in the DLD group to rare changes compared with the TD group. Additionally, if groups differed in both types of blocks, we could conclude that children with DLD have a more general deficit in combining auditory and visual stimuli. On the other hand, if groups differed only in blocks with the articulating mouth, the data would point to a more specific deficit in incorporating visual features into long-term phonemic representations.

## 2. Materials and Methods

### 2.1. Participants

A total of 18 children with TD (7 female) and 18 children with DLD (5 female) participated in the study. The mean age for children with TD was 10 years and 2 months (range from 7 years, 3 months to 13 years, 7 months). The mean age for children with DLD was 10 years (range from 7 years, 7 months to 13 years, 8 months). Two children in the TD group and 3 children in the DLD group were left-handed. The rest were right-handed. All gave their written assent to participate in the experiment. Additionally, at least one parent of each child has provided a written consent for participating in the study. The study was approved by the Institutional Review Board of Purdue University (protocol # 0909008484), and all study procedures conformed to The Code of Ethics of the World Medical Association [61].

### 2.2. Screening Measures

Children with DLD were originally diagnosed with SLI during preschool years (between 3 years, 11 months and 5 years, 9 months of age) based on either the Structured Photographic Expressive Language Test—2nd Edition (SPELT-II, [62]) or the Structured Photographic Expressive Language Test—Preschool 2 (SPELT-P2; [63]). These tests have shown good sensitivity and specificity [64,65]. Children diagnosed with SPELT-P2 (n = 13) received the standard score of 86 or less (mean 76, range 61–86, SD = 8.4). According to the study by Greenslade and colleagues [64], the cut-off point of 87 provides good sensitivity and specificity for the tested age range. All children’s standard scores on SPELT-P2 fell below the 24th percentile (mean 10.5, range 2–23, SD = 6.9). Children diagnosed with SPELT-II (n = 5) received raw scores of 18–26, all of which fell below the 5th percentile. In sum, the children with DLD showed significant language impairment at the time of the diagnosis. All but one of them had received some form of language therapy in the years between the original diagnosis of DLD and the current study (mean of 4.9 years, range 1–9 years, SD = 2.2), with 7 children still receiving therapy at the time of this study. Most of these children participated in multiple unrelated studies in our lab. Results from several of these concurrent studies have been published before [14,66]. Each study had a unique design and focused on a specific audiovisual skill. Screening tests, as described below, were administered once, after which children participated in several electroencephalographic (EEG) recording sessions.

We administered 4 subtests of the Clinical Evaluation of Language Fundamentals (CELF-4) to all children in order to assess their current language ability—the Concepts and Following Directions (C&FD, 7–12-year-olds only), Recalling Sentences (RS), Formulated Sentences (FS), Word Structure (WS, 7 and 8-year-olds only), Word Classes-2 Total (WC-2, 9–12-year-olds only), and Word Definitions (WD, 13-year-olds only). Taken together, these subtests yielded the Core Language Score (CLS), which reflects general linguistic aptitude and the potential presence of language impairment. Additionally, we evaluated children’s verbal working memory with a nonword repetition test [67] and the Number Memory Forward and Number Memory Reversed subtests of the Test of Auditory Processing Skills—3rd edition (TAPS-3; [68]). All children were administered the Test of Nonverbal Intelligence—4th edition (TONI-4; [69]) to rule out intellectual disability and the Childhood Autism Rating Scale—2nd edition [70] to rule out the presence of autism spectrum disorders. The level of mothers’ and fathers’ education was measured as an indicator of children’s socio-economic status (SES). The level of risk for developing attention deficit/hyperactivity disorder (ADHD) was evaluated with the help of the short version of the Parent Rating Scale of the Conners’ Rating Scales—Revised [71]. In all participants, handedness was assessed with an augmented version of the Edinburgh Handedness Questionnaire [72,73].

Eight children with DLD had a current diagnosis of attention deficit/hyperactivity disorder (ADHD), with 5 taking medications to control symptoms. The presence of attention and working memory problems in children with DLD is common [18]. Because ADHD is highly co-morbid with DLD [74] and because language difficulties associated with ADHD proper are at least partially different from the language difficulties associated with DLD [75,76], we did not exclude these children from our sample. None of the TD children had any history of atypical language development, ADHD, or reading difficulties. All participants were free of neurological disorders (e.g., seizures), passed a hearing screening at a level of 20 dB HL at 500, 1000, 2000, 3000, and 4000 Hz, and reported to have normal or corrected-to-normal vision.

### 2.3. Design

The visual stimuli consisted of still images of a speaker producing a vowel [i] as in “beet,” a vowel [u] as in “boot,” and with a mouth closed and in a neutral position. The auditory stimuli consisted of vowels [i] and [u]. The frequency with which each of these sounds were heard made two types of blocks that were used. In one, the sound [i] occurred frequently (i.e., was a standard) while the sound [u] occurred infrequently (i.e., was a deviant). In the other, the reverse was true. The study had 2 conditions, which differed in how visual stimuli were paired within each block. A schematic representation of both conditions is shown in Figure 1. In neutral condition, the image on the screen was always of a face with a closed mouth. In this condition, standards were consistent pairings between auditory and visual stimuli, but the shape of the speaker’s lips was neutral and never matched the heard vowels. We, therefore, expected that the neural response to deviants in this condition would reflect a violation of the observed audiovisual pattern but would not tap into long-term audiovisual representations of phonemes. In the audiovisual violation condition, the same two types of blocks were used; however, the image of the speaker always matched the frequent vowel. For example, in blocks where [i] was a standard, the image of the speaker with a narrow and long mouth opening was shown. Because in standards the speaker’s mouth shape matched the frequently heard vowel, we expected that deviants in this condition violated not only observed audiovisual patterns but also the long-term audiovisual representations for phonemes [i] and [u] and should result in a larger amplitude of MMN. The ratio of standards to deviants in all blocks was 80/20.

In order to make the design appropriate for children, the speaker was wearing a wolf costume. The instructions provided to children incorporated the wolf character. Children were told that the wolf helped researchers record videos, but occasionally he “goofed off” and did something silly instead. Children were asked to help researchers identify all these silly instances by closely watching the videos and by pressing a response button as soon as they saw a wolf stick a tongue out, make a fish face, or growl. The number of these silly events was equal to the number of deviants in each block. We recorded children’s accuracy and response time when detecting silly events. However, our main measures were electrophysiological deflections associated with deviance detections—namely, the MMN and P3 components. Each block consisted of 72 trials: 48 standards, 12 deviants, and 12 silly faces. All 3 types of trials (standards, deviants, and silly faces) were presented pseudo-randomly, with 2 consecutive standards starting each block and with no more than 1 deviant in a row. Each child completed 16 blocks of testing (8 blocks of the neutral condition and 8 blocks of audiovisual violation condition), with game breaks after each block. The 16 blocks of the experiment were randomized for each participant.

### 2.4. ERP Measures

The electroencephalographic (EEG) data were recorded from the scalp at a sampling rate of 512 Hz using 32 active Ag–AgCl electrodes secured in an elastic cap (Electro-Cap International Inc., USA). Electrodes were positioned over homologous locations across the two hemispheres according to the criteria of the International 10–10 system [62]. The specific locations were as follows: midline sites Fz, Cz, Pz, and Oz; mid-lateral sites FP1/FP2, AF3/AF4, F3/F4, FC1/FC2, C3/C4, CP1/CP2, P3/P4, PO3/PO4, and O1/O2; lateral sites F7/F8, FC5/FC6, T7/T8, CP5/CP6, and P7/P8, and left and right mastoids. EEG recordings were made with the Active-Two System (BioSemi Instrumentation, Netherlands), in which the Common Mode Sense (CMS) active electrode and the Driven Right Leg (DRL) passive electrode replace the traditional “ground” electrode [63]. Data were referenced offline to the electrode positioned at the tip of the nose (Nz). The Active-Two System allows EEG recording with high impedances by amplifying the signal directly at the electrode [64,65]. In order to monitor for eye movement, additional electrodes were placed over the right and left outer canthi (horizontal eye movement) and below the left eye (vertical eye movement). Prior to data analysis, EEG recordings were filtered between 0.1 and 30 Hz. Individual EEG records were visually inspected to exclude trials containing excessive muscular and other non-ocular artifacts. Ocular artifacts were corrected by applying a spatial filter (EMSE Data Editor, Source Signal Imaging Inc., La Mesa, CA, USA) [66]. ERPs were epoched starting at 200 ms pre-stimulus and ending at 1000 ms post-stimulus onset. The 200 ms prior to the stimulus onset served as a baseline. The number of useable trials for each group, condition (neutral or audiovisual violation), and stimulus type (standard or deviant) following eye artifact correction and exclusions is shown in Appendix A. There were no differences between conditions, *F*(1,34) = 2, *p* = 0.17, and *ƞ_p_^2^* = 0.06, or groups, *F*(1,34) = 2.08, *p* = 0.16, and *ƞ_p_^2^* = 0.06, and none of the relevant interactions were significant: condition by group, *F*(1,34) = 0.004, *p* = 0.95, and *ƞ_p_^2 =^* 0.000; stimulus type by group, *F*(1,34) = 1.42, *p* = 0.24, and *ƞ_p_^2^* = 0.04; condition by stimulus type, *F*(1,34) = 0.92, *p* = 0.34, and *ƞ_p_^2^* = 0.03.

The MMN and P3 ERP components were measured as differences in the mean amplitude between deviants and standards 150 and 250 ms (MMN) and 250 and 350 ms (P3) post sound onset. The MMN component was measured over two sets of electrodes—the frontal and central sites (AF3, F3, FC1, and C3 over the left hemisphere and AF4, F4, FC2, and C4 over the right hemisphere) and over the parietal and occipital sites (CP1, P3, PO1, and O1 over the left hemisphere and CP2, P4, PO2, and O2 over the right hemisphere). These measures are referred to henceforth as anterior and posterior MMN. The P3 component was measured over the frontal and central sites (F3, FC1, and C3 over the left hemisphere and F4, FC2, and C4 over the right hemisphere). The standard that started each block as well as the first standard after each deviant and silly expression were excluded from averages and analyses.

### 2.5. Statistical Analysis

One-way ANOVA was used to compare groups’ means on measures of language, nonverbal intelligence, working memory, autism traits, risk for ADHD, age, and parental education.

A mixed-effects model was used to model response time and accuracy separately as a function of group (DLD vs. TD), condition (neutral vs. audiovisual violation), and group-by-condition interaction. Simple effects are estimated for this interaction. A random intercept was included to account for the repeated measures nested within subjects.

Similarly, for the anterior MMN, posterior MMN, and P3 ERP components, a series of mixed-effects models were used to model mean amplitude differences as a function of group, condition, hemisphere (left, midline, and right), and electrode site. First, a main effects model was estimated to obtain main effects of the independent variables. Post-model contrasts were used to test independent variables with more than two categories, that is, hemisphere. Next, a model with all possible 2- and 3-way interactions between group, condition, and hemisphere was estimated controlling for site. Finally, a best (most parsimonious) model was chosen by removing interactions falling below alpha = 0.05. Simple effects for interactions that remain in the best model were estimated. Results from both the main effects and best models are reported. Both unstandardized (b) and partially standardized regression coefficient estimates (b_std_) where the outcome is standardized are provided. Multiple testing is accommodated in the mixed-effects model estimation [67].

## 3. Results

### 3.1. Screening Measures

Table 1 and Table 2 summarize children’s performance on all screening tests. As expected, the groups did not differ either in nonverbal intelligence or the presence of behaviors suggestive of autism. Mothers in both groups were well matched on the level of education, but fathers of children with DLD had on average four fewer years of formal education, which was a significant difference. Children with DLD performed significantly worse on almost all language, working memory, and attention measures. Nonword repetition had been shown to be particularly sensitive to language difficulties in children with DLD for a meta-analysis of studies [68]. In our sample, children with DLD performed significantly worse than their peers with TD on three- and four-syllable nonwords and marginally worse on two-syllable nonwords. Taken together, these findings agree with earlier research showing that difficulty with language and other cognitive skills among children with DLD persists into the school age and beyond (e.g., [22,23,69,70,71,72]).

### 3.2. Behavioral and ERP Results

Table 3 shows children’s accuracy and response times in detecting silly facial expressions. Appendix A provide estimates from statistical models of accuracy and response times, respectively. Table 4 and Table 5 provide simple effects tests from the models in Appendix A, respectively.

In absolute terms, children with TD were slightly more accurate than children with DLD. Although this group difference did not reach statistical significance (b_std_ = −0.60, *p* = 0.06; Appendix A), the effect size was large. There was no effect of condition (b_std_ = −0.10, *p* = 0.23; Appendix A) and no condition-by-group interaction (*p* = 0.65). The overall high accuracy in both groups (over 95% of silly expressions detected) suggests that children were able to keep their attention on the computer monitor and perform the task.

The groups did not differ in overall response time (b_std_ = −0.26, *p* = 0.43; Appendix A). Responses were slower in the audiovisual violation condition, (b_std_ = 11.95, *p* = 0.003; Appendix A). The effect of condition did not interact with group (*p* = 0.06; Appendix A). The examination of the simple effects in each group (see Table 5) suggested that the effect of condition was driven primarily by the DLD group (b_std_ = 0.20, *p* = 0.001).

ERPs elicited by standards and deviants over the posterior and anterior scalp are shown in Figure 2 and Figure 3, respectively. The difference waveforms resulting from subtracting ERPs elicited by standards from the ERPs elicited by deviants are shown in Figure 4 for both groups and conditions. The distribution of voltage in the MMN time window over the entire scalp is shown in Figure 5.

Mixed-model results for the *posterior MMN* component are presented in Appendix A, and the simple effects results from the interaction model are shown in Table 6. There was a main effect of condition, (b_std_ = −0.35, *p* < 0.001; Appendix A), with larger amplitude in the AV violation condition. There was no main effect of group (*p* = 0.24; Appendix A), but there was a group-by-condition interaction (b_std_ = 0.50, *p* < 0.001; Appendix A). The MMN mean amplitude over the left hemisphere sites was smaller than over the right hemisphere sites (b_std_ = −0.45, *p* = 0.01; Appendix A).

The simple effects (Table 6) stemming from the group-by-condition interaction revealed that there was a condition effect for the TD group (b_std_ = −0.61, *p* < 0.001), with larger MMN in the AV violation compared with the neutral condition, but not for the DLD group (b_std_ = −0.10, *p* = 0.17). In addition, there was a group difference in the AV violation condition (b_std_ = 0.52, *p* = 0.025), with larger amplitude in the TD group, but not in the neutral condition (*p* = 0.96). These group differences are shown in a bar graph format in Figure 6.

Results of the *anterior MMN* analysis were parallel to those of the posterior MMN described above although with smaller effect sizes (see Appendix A and Table 7). There was again no main effect of group (*p* = 0.27; Appendix A). There was a main effect of condition (b_std_ = −0.16, *p* = 0.008; Appendix A), with less positive voltage in the AV violation condition (note that when pooled across both groups, the MMN measurement was positive, which means that the MMN was absent at this level of analysis), and there was a group-by-condition interaction (b_std_ = 0.41, *p* = 0.001; Appendix A). The simple effects (Table 7) stemming from the group-by-condition interaction revealed that the condition effect (with larger MMN in the AV violation condition) was evident in children with TD (b_std_ = −0.37, *p* < 0.001) but not in children with DLD (*p* = 0.64) and, again, the DLD group differed from the TD group only in the AV violation condition (b_std_ = 0.41, *p* = 0.04), with larger MMN amplitude in children with TD. The main effect of hemisphere was also present, with the MMN measurement being less positive over the right as compared to the left (b_std_ = −0.16, *p* = 0.02; Appendix A) and midline (b_std_ = −0.37, *p* < 0.001; Appendix A) sites and less positive over the left compared to the midline sites (b_std_ = −0.21, *p* = 0.01; Appendix A).

Finally, the *P3 mean amplitude* was analyzed (see Appendix A). The main effects of condition (*p* = 0.27) and group (*p* = 0.68) were not statistically significant, and there was no condition-by-group interaction (*p* = 0.31). The P3 mean amplitude was smaller over the right hemisphere sites compared with over the left hemisphere (b_std_ = −0.28, *p* < 0.001) and midline (b_std_ = −0.38, *p* < 0.001) sites.

In summary, both groups of children performed the secondary task of detecting silly facial expressions with high accuracy, suggesting that they were able to maintain their attention on the monitor. Most importantly, in agreement with our hypothesis, deviants in the audiovisual violation condition elicited a larger MMN than deviants in the neutral condition in the children with TD only.

### 3.3. Regressions

To determine whether the amplitude of posterior MMN correlated with children’s language proficiency as measured by CELF-4 and/or the phonological short-term memory as measured by nonword repetition, we conducted bivariate correlations between these variables with data from both groups of children combined. Neither relationship was significant: language proficiency, *r* = −0.23, *p* (two-tailed) = 0.19; accuracy of repeating four-syllable nonwords, *r* = −0.183, *p* (two-tailed) = 28.

## 4. Discussion

In this study, we set out to determine whether long-term phonemic representations in children with DLD incorporate visual information to the same extent as in children with TD. The ERP results reported above showed that only children with TD were sensitive to audiovisual patterns that violated a typical combination of heard vowel and seen mouth shape. This finding is significant because it shows that phonological deficits in DLD are not limited to the auditory modality, but, rather, are multisensory in nature.

Two aspects of the reported results deserve additional emphasis. First, the voltage maxima of the MMN elicited by the audiovisual violation condition in children with TD was over the posterior scalp, suggestive of the visual cortex involvement, even though standards and deviants differed only auditorily. This scalp distribution lends further support to the suggestion that auditory changes present in deviants were perceived as disruptions in audiovisual rather than in auditory-only sequences.

Second, the absence of MMN in the neutral condition in both groups of children was unexpected. One possible explanation for this finding is that incongruency between auditory and visual stimuli in this condition has precluded children from integrating the two modalities into an audiovisual percept. If so, rare sounds changes would have been expected to elicit MMN with fronto-central distribution typical for auditory oddball paradigms. A prominent P3 component over the anterior scalp present in both groups and conditions might have obscured such an MMN.

Indeed, the presence of P3 in both groups of children suggests that they did notice the auditory change and that their attention was drawn to deviants. However, the P3 amplitude did not differ between conditions and groups. The presence of the P3 component in the ERPs of children with DLD in the absence of the MMN component may seem surprising, but it fits well with earlier studies on the cognitive mechanisms eliciting MMN. The presence of MMN is heavily context-dependent. It can, therefore, be absent if the deviant sound/feature/object, etc., is not part of the formed standard against which a deviant is compared [73]. In other words, if visual information was not part of the perceived regularity for children with DLD, the auditory deviant that violated audiovisual congruency did not signal a change in pattern and did not elicit MMN but did capture attention and elicited P3.

The consequences of not encoding visual features into long-term phonemic representations for language development more generally and for children with DLD more specifically will require future studies that examine different domains of linguistic competence in this group in greater detail. As described in the Introduction, studies of early language acquisition have begun to reveal that infants are attuned to visual components of speech from very early in development and that this sensitivity is subject to developmental changes similar to those in the auditory domain. If children with DLD are less sensitive to audiovisual correspondences in infancy—a proposition that is, admittedly, speculative at the moment—one might expect less robust phonological encoding in this group of children, especially in the presence of background noise, which is often the norm in daycare facilities. Previous studies show that children [74] and infants [75] benefit from seeing the talker’s face when listening to speech in a noisy environment. Importantly, children with DLD have been shown to struggle significantly more than their peers with TD when listening to speech-in-noise (e.g., [14,76,77]). In fact, in the study by Ziegler and colleagues [78], children could be reliably identified as either language impaired or not based on their performance on the speech-in-noise perception task. It is also noteworthy that robust multisensory processing may impart benefits not only at the time of encoding but also during later *unisensory* perception. This beneficial influence of multisensory stimuli has been documented in both linguistic and nonlinguistic domains (e.g., [79,80]).

Another possible consequence of atypical audiovisual processing at the phonemic level may be delayed lexical development. Several studies have reported that infants who prefer to fixate the mouth, rather than the eyes, of the talker, tend to have higher expressive language skills either concurrently [81] or during subsequent years of life [82,83]. This relationship makes a lot of sense given that phonological and lexical development proceed in parallel and support each other [84]. Studies of lexical acquisition in children with DLD show that these children acquire their first words significantly later than children with TD [85] and phonological representations of words in this group appear to be less detailed (e.g., [86]).

Despite the relationship between face fixation preferences and language development reported above, in our own study the correlation between the size of the posterior MMN component and children’s general linguistic ability and phonological short-term memory (as measured by a nonword repetition task) was not significant. It is possible that the measure of language skills reflected in the CELF-4 score was too broad to correlate with a much more specific electrophysiological measure. Additionally, the relationship between audiovisual encoding and linguistic abilities is likely to be age-dependent, with perhaps greater influence of other cognitive functions on this relationship in older children. Understanding connections between specific audiovisual skills and specific language functions will require a great deal more work but promises to provide a new perspective on typical and impaired language use.

## 5. Limitations and Future Directions

The current study consisted of only audiovisual conditions, which was mainly due to the need to limit the length of experimental sessions for children. Additional conditions, especially the one with video-only changes, could further clarify the nature of audiovisual deficits in children with DLD. One of our earlier studies reported a reduced visual N1 to the onset of human faces in this population [14]. Therefore, it is possible that the encoding of visual features relevant for speech perception, and not just their integration with the auditory signal, is also atypical in children with DLD. This hypothesis is currently being tested by ongoing studies in our laboratory. Children with DLD are a heterogeneous group based on the range of observed impairments and the range of co-occurring disorders. In agreement with the current view of this disorder in the field of speech−language pathology [18], we have included children with ADD/ADHD in the DLD group because these disorders frequently co-occur. However, this may have increased the overall variability in our DLD sample. Finally, relatively few studies had previously examined the conditions under which the audiovisual MMN may be elicited and whether its neural correlates are different from those of auditory and visual MMN components. More studies focusing on the properties of audiovisual MMN are needed in order to improve our understanding of its significance.

## Figures and Tables

**Figure 1 brainsci-11-00507-f001:**
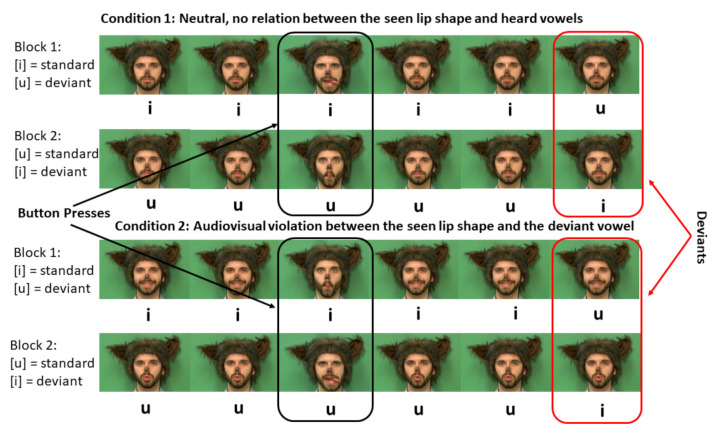
Experimental Design. Note that children pressed a response button every time they saw the wolf do something silly. No responses were provided for stimuli that served as experimental deviants.

**Figure 2 brainsci-11-00507-f002:**
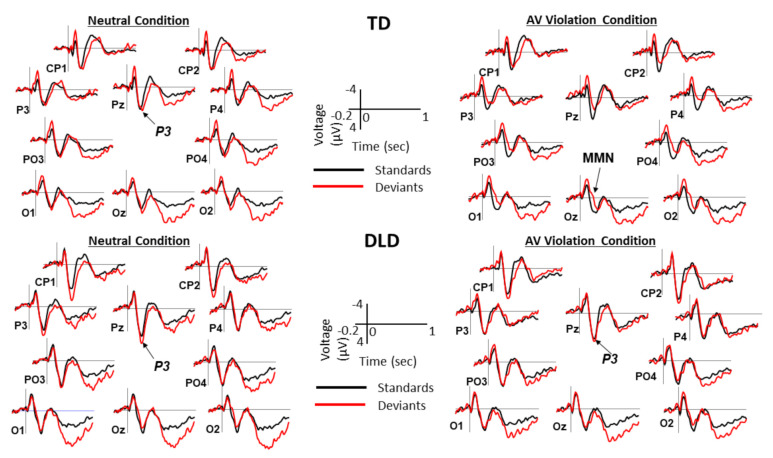
Event-related potential (ERPs) elicited by standards and deviants over the posterior scalp. Grand average ERPs over the 12 posterior sites are shown separately for children with TD and children with DLD. In each condition, grand averages for standards (black lines) are overlapped with grand averages for deviants (red lines). Negative is plotted up. The P3 and MMN components are marked on the Pz and Oz sites, respectively, where present. Time 0 marks the onset of sound. AV = audiovisual.

**Figure 3 brainsci-11-00507-f003:**
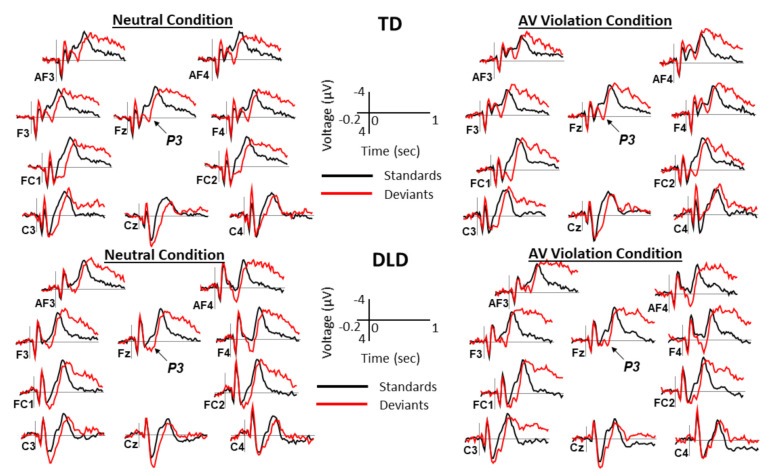
ERPs elicited by standards and deviants over the anterior scalp. Grand average ERPs over the 12 anterior sites are shown separately for children with TD and those with DLD. In each condition, grand averages for standards (black lines) are overlapped with grand averages for deviants (red lines). Negative is plotted up. The P3 component is marked on the Fz site. Time 0 marks the onset of sound. AV = audiovisual.

**Figure 4 brainsci-11-00507-f004:**
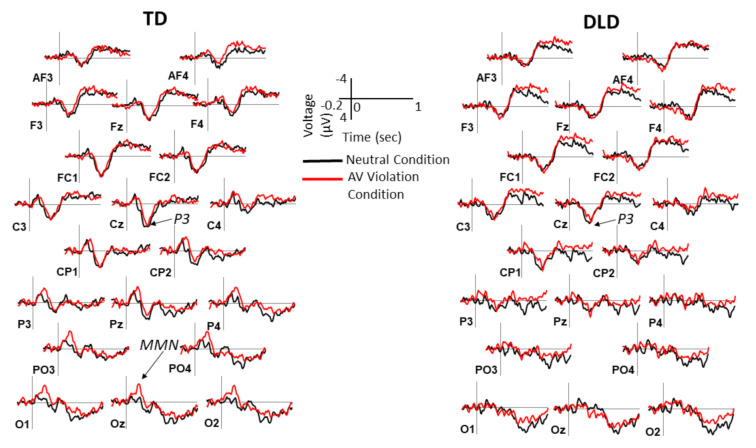
Deviant minus standard difference waveforms in neutral and audiovisual violation conditions in each group. Grand average ERPs for deviant minus standard differences are shown separately for children with TD and children with DLD. Note the prominent MMN component posteriorly in the audiovisual violation condition in children with TD and its absence in children with DLD. Negative is plotted up. The P3 component is marked on the Cz site and the MMN component on the Oz site (where present). Time 0 marks the onset of sound. AV = audiovisual.

**Figure 5 brainsci-11-00507-f005:**
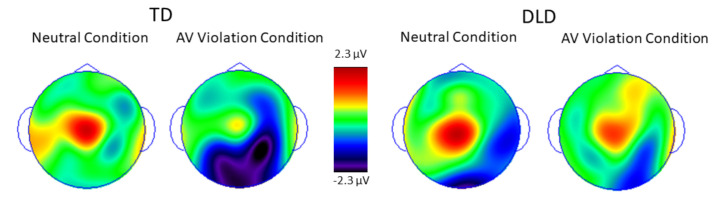
Voltage distribution 150–250 ms post-stimulus onset in deviant—standard difference waves. Note the significant negative voltage over posterior scalp in the AV violation condition in children with TD but not in children with DLD.

**Figure 6 brainsci-11-00507-f006:**
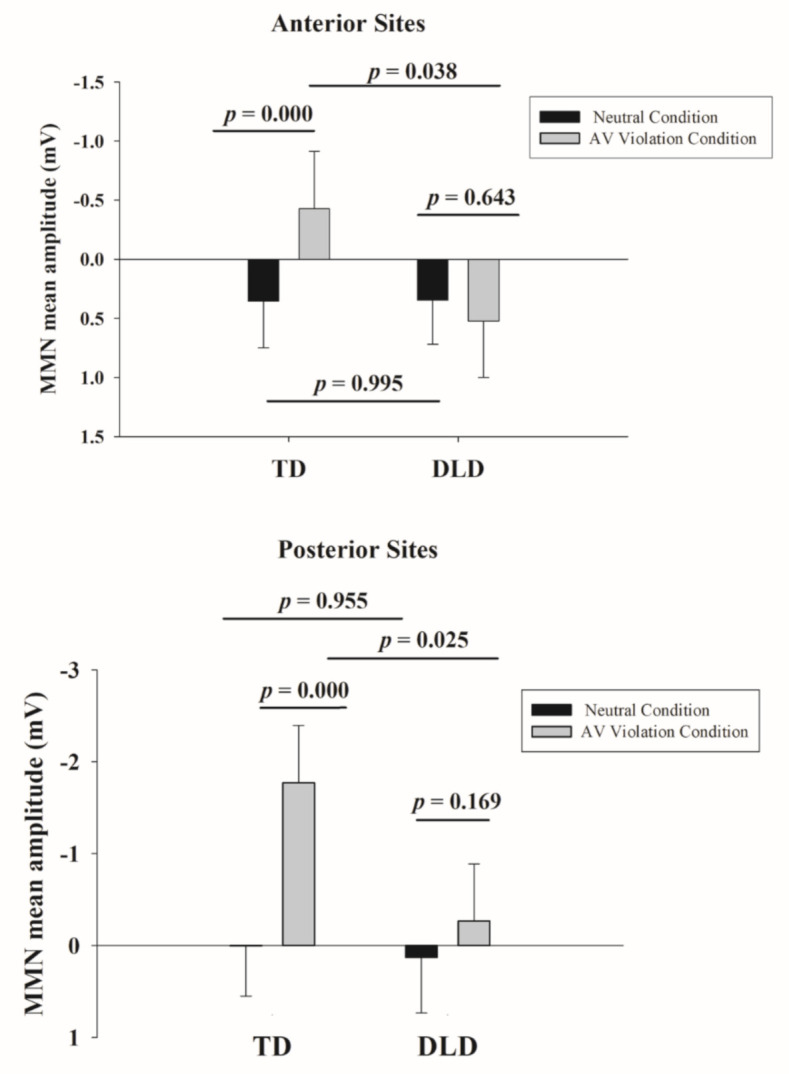
Mismatch negativity (MMN) mean amplitude.

**Table 1 brainsci-11-00507-t001:** Group means for age, nonverbal intelligence (Test of Nonverbal Intelligence—4th edition, TONI-4), presence of autism (Childhood Autism Rating Scale, 2nd edition, CARS-2), socio-economic status (parents’ education level), and linguistic ability (Clinical Evaluation of Language Fundamentals, 4th edition, CELF-4).

	Age (Years, Months)	TONI-4	CARS	Mother’s Education (Years)	Father’s Education (Years)	CELF-4					
						CFD	RS	FS	WS	WC2 R/E/T	CLS
DLD	10; 1 (0.4)	105.8 (1.7)	15.6 (0.3)	15.2 (0.8)	13.5 (0.6)	9.4 (0.5)	7.7 (0.6)	9.9 (0.4)	9.4 (1.3)	11.0 (0.8)/10.1 (0.7)/10.5 (0.7)	95.8 (2.5)
TD	10; 1 (0.4)	109.0 (2.4)	15.1 (0.1)	15.7 (0.5)	17.2 (0.8)	11.8 (0.5)	11.8 (0.5)	12.7 (0.4)	11.0 (0.4)	13.6 (0.6)/12.0 (0.5)/13.0 (0.5)	113.7 (2.2)
*F*	<1	1.11	3.39	<1	13.58	10.23	26.92	23.39	2.177	7.26/5.72/7.65	28.12
*p*	0.96	0.3	0.08	0.6	0.001	0.003	<0.001	<0.001	0.3	0.01/0.03/0.01	<0.001

Note. Numbers for TONI-4, CARS-2, and the CELF-4 subtests represent standardized scores. Numbers in parentheses are standard errors of the mean. P and F values reflect a group comparison. When the homogeneity of variance differed between groups, the Brown–Forsythe robust test of equality of means was used to determine significance, and the corresponding *p*-values are reported. CFD = concepts and following directions; RS = recalling sentences; FS = formulated sentences; WS = word structure; WC2 = word classes; R = receptive; E = expressive; T = total; CLS = core language score; DLD = developmental language disorder; TD = typical development.

**Table 2 brainsci-11-00507-t002:** Group means for nonword repetition, auditory processing (Test of Auditory Processing Skills—3rd edition, TAPS-3), and attention deficit/hyperactivity disorder (ADHD) symptoms (Conners’ Rating Scales).

	Nonword Repetition				TAPS-3		Conners’
	Syllables				Number Memory		
	1	2	3	4	Forward	Reversed	ADHD Index
DLD	99.5 (0.5)	94.2 (2.0)	89.6 (2.0)	67.7 (3.6)	7.4 (0.5)	9 (0.4)	55.8 (2.6)
TD	100 (0.0)	98.6 (0.7)	98 (0.7)	84.9 (2.4)	10.8 (0.6)	11.8 (0.7)	47.9 (1.3)
*F*	1.0	4.3	15.1	15.7	20.1	11.8	6.629
*p*	0.32	0.05	0.001	<0.001	<0.001	0.002	0.012

Note. Numbers for TAPS-3 and Conners’ represent standardized scores. Numbers for nonword repetition reflect percent correct of repeated phonemes. Numbers in parentheses are standard errors of the mean. When the homogeneity of variance differed between groups, the Brown–Forsythe robust test of equality of means was used to determine significance, and the corresponding *p*-values are reported.

**Table 3 brainsci-11-00507-t003:** Group means for accuracy (ACC) and response time (RT) while detecting silly facial expressions.

	ACC (% Correct)		RT (ms)	
	Neutral Face	Articulating Face	Neutral Face	Articulating Face
DLD	95.4 (1.2)	95.2 (1.1)	594.5 (21.2)	613.7 (21.6)
TD	97.9 (0.4)	97.4 (0.6)	626.5 (24.3)	631.2 (22.1)

Note: Numbers in parentheses are standard errors of the mean.

**Table 4 brainsci-11-00507-t004:** Accuracy model simple effects for the group-by-condition interaction.

	b	*p*-Value	95% CI		b_std_
DLD versus TD neutral condition	−2.49	0.050	−4.98	0.00	−0.64
DLD versus TD AV violation condition	−2.20	0.083	−4.69	0.29	−0.56
AV violation vs. neutral for DLD group	−0.23	0.606	−1.11	0.65	−0.06
AV violation vs. neutral for TD group	−0.52	0.246	−1.40	0.36	−0.13

Note. b_std_ is a partially standardized coefficient where the outcome is standardized, comparable to a conditional. Cohen’s d. AV = audiovisual.

**Table 5 brainsci-11-00507-t005:** Response time model simple effects for the group-by-condition interaction.

	b	*p*-Value	95% CI		b_std_
DLD versus TD neutral condition	−32.00	0.31	−93.84	29.84	−0.34
DLD versus TD AV violation condition	−17.48	0.58	−79.32	44.36	−0.19
AV violation vs. neutral for DLD group	19.21	0.001	8.36	30.06	0.20
AV violation vs. neutral for TD group	4.69	0.397	−6.16	15.54	0.05

Note. b_std_ is a partially standardized coefficient where the outcome is standardized, comparable to a conditional. Cohen’s d.

**Table 6 brainsci-11-00507-t006:** Posterior MMN model simple effects for the group-by-condition interaction.

	b	*p*-Value	95% CI		b_std_
DLD versus TD neutral condition	0.04	0.955	−1.31	1.38	0.01
DLD versus TD AV violation condition	1.53	0.025	0.19	2.88	0.52
AV violation vs. neutral for DLD group	−0.30	0.169	−0.73	0.13	−0.10
AV violation vs. neutral for TD group	−1.80	0.000	−2.23	−1.37	−0.61

**Table 7 brainsci-11-00507-t007:** Anterior MMN model simple effects for the group-by-condition interaction.

	b	*p*-Value	95% CI		b_std_
DLD versus TD neutral condition	0.00	0.995	−0.86	0.87	0.00
DLD versus TD AV violation condition	0.92	0.038	0.05	1.78	0.41
AV violation vs. neutral for DLD group	0.09	0.643	−0.29	0.47	0.04
AV violation vs. neutral for TD group	−0.82	0.000	−1.20	−0.44	−0.37

## Data Availability

Data available from the corresponding offer upon a reasonable request.

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
