# Peer review of "Impaired Audiovisual Representation of Phonemes in Children with Developmental Language Disorder"

_brainsci, 2021, doi:10.3390/brainsci11040507_

Round 1
Reviewer 1 Report
I enjoyed reading this original study. Please see the attachment

Reviewer 2 Report
This study used 32-channel EEG to examine the audiovisual representation of phonemes in children with developmental language disorders (DLD) compared to their typically developing peers (TD). They found that only children with TD were sensitive to audiovisual patterns that violated a typical combination of heard vowel and seen mouth shape, but not children with DLD. Thus, they concluded that the phonological deficits in DLD are not limited to the auditory modality, but rather, are multisensory in nature. This is an interesting study but it needs minor revision before it can be accepted for publication. The detailed comments are listed as follows:
1. Introduction
The first two paragraphs are well written for the brain regions for audio-visual speech tasks. However, they seem to be irrelevant to the EEG data presented here. The 32-channel EEG data collected in the current study can't add meaningful spatial information about the brain regions involved in the audio-visual representation of phonemes. Recommendation: start with EEG literature on posterior and anterior scalp data in children with similar paradigms. Or start with the third paragraph in the current introduction.
What is the importance of audiovisual integration in language development? It is not clearly stated in the introduction. What we learned from this study can help children with DLD?
Why Phonemes are the focuses of this study? Are children with a mean age of 10 years old ceiling out on phoneme task?
Face expression can elicit different brain activations. Why the paradigm uses distracted faces instead of simple mouth only pictures? Any supported rationale for the specific choice of the face used in this study?
Studies of audiovisual MMN are few. Only one group's results are presented. The sensitivity of audiovisual MMN can be questionable. The authors may want to address this in the limitation section of the discussion.
2. Materials and Methods
8 out of 18 children with DLD had ADHD. This is a limitation for the current study to generalize their results to the population of children with DLD. Please address this in the limitation section of the discussion.
60 trials/block, 16 blocks, Each child finished 960 trials. How many trials for each condition (standards, deviants, and silly faces) were presented in a block? 20 trials for each condition? How long does the whole experiment take?
2.4 ERP measures
F tests for conditions, ηp2? What does this value mean? Table 1 can be put into supplementary information.
2.5 Statistical Analysis
There are numerous comparisons among groups, etc.. How did the authors handle multiple comparison corrections?
3. Results
It is overwhelming to see so many tables. Some of them can be put into supplementary information.
3.2 Behavioral and ERP results
Table 4, can the authors add statistical comparison results between DLD and TD groups? as well as for Neutral vs. Articulating Face?
The table 5 can be confusing. Main effects and interaction model, are these tested separately or together? It is straightforward to present the statistics in Table 4 instead of using another table 5. Interaction is not significant. Then, a two-sample t-test can be used for between-group comparison on ACC and RT data. A paired t-test can be used for within-group comparison on two conditions.
Table 5a, 6, and 6a are not needed if the authors take the suggestions. If the authors want to keep these results, recommend putting them in the supplementary information. They are not necessary to be included in the results. Table 5, 5a, and 6, 6a can be incorporated into Table 4.
Figure 2 and 3: It will be better if the group data were put on the same graph with different colors of lines so that the reader knows they are on the same scale. But it is not critical to put the group results on one graph. If the authors only focus on Pz Oz for the posterior scalp. Figure 2 can be simplified to show Pz and Oz waveforms. It will make the results clear and neat. The rest can be put into supplementary information. Same for Figure 3, if Fz is the focus, the rest can be put into supplementary information.
Same for Figure 4, if Cz and Oz are the focuses, the rest can be put into supplementary information.
Table 7, if the group x condition interaction term is significant as stated in the table, the two group data should be analyzed separately. Same for Table 8. Too many tables in the paper. Please consider condensing or put some of them into supplementary information. The description of the results covered the information presented in the table. No need to put the tables in the main text.
Table 9 can be put into supplementary information as well.
3.3 Regressions
Are the two groups separated for this analysis since there is significant interaction between group and condition?
4. Discussion
Paragraph 5 talked about the developmental aspect. But this study really can comment on the developmental aspect since it only studied children with a mean age of 10.
Please add a section to state the limitation of this study. There are significant limitations that need to be clearly stated for this study.
